# A Predicted Mannoprotein Cmp1 Regulates Fungal Virulence in *Cryptococcus neoformans*

**DOI:** 10.3390/pathogens9110881

**Published:** 2020-10-24

**Authors:** Lian-Tao Han, Lei Wu, Tong-Bao Liu

**Affiliations:** 1State Key Laboratory of Silkworm Genome Biology, Southwest University, Chongqing 400715, China; hlt892996713@email.swu.edu.cn (L.-T.H.); 18215522009@163.com (L.W.); 2Chongqing Key Laboratory of Microsporidia Infection and Control, Southwest University, Chongqing 400715, China

**Keywords:** *Cryptococcus neoforma*ns, mannoprotein, Cmp1, virulence, capsule

## Abstract

The capsule of the fungal pathogen *Cryptococcus neoformans* consists of glucuronoxylomannan (GXM), glucuronoxylomannogalactan (GXMGal), and mannoproteins (MPs). MPs are a kind of glycoproteins with low content but high immunogenicity, which can stimulate the immune protection of the host. However, there is not much information about the role of mannoproteins in virulence of the human fungal pathogen *C. neoformans*. In this study, we reported the identification and functional analysis of a predicted mannoprotein Cmp1 that regulates fungal virulence in *C. neoformans*. Gene expression pattern analysis indicates that the *CMP1* gene was ubiquitously expressed at all stages of cryptococcal development. Subcellular localization analysis indicated that Cmp1 was localized in the cytoplasm of cryptococcal cells. Disruption or overexpression of *CMP1* results in impairing capsule formation in *Cryptococ*cus, but it does not affect the melanin production and sensitivity under various stress conditions, nor does it affect the sexual reproduction process of *Cryptococcus*. Survival assay showed that the pathogenicity of the *cmp1*Δ mutant or the *CMP1* overexpression strain was significantly attenuated in a murine inhalation model of cryptococcosis. In conclusion, our findings implied that the mannoprotein Cmp1 is required for the virulence of *C. neoformans*.

## 1. Introduction

*Cryptococcus neoformans* is an encapsulated yeast-like pathogen that can cross the blood–brain barrier to cause fungal meningitis predominantly in immunocompromised patients, leading to hundreds of thousands of deaths each year [1,2,3]. As a human pathogenic fungus, *C. neoformans* possess several virulence factors such as capsule and melanin, and growth at 37 °C, which contribute to the survival of *C. neoformans* in the host [4,5,6]. Among the virulence factors, polysaccharide capsules are one of the most crucial factors to cause disease [7,8]. The capsule is comprised of glucuronoxylomannan (GXM, 90%), glucuronoxylomannogalactan (GXMGal, 9%), and small proportions of mannoproteins (MPs, 1%) [7,9].

There are many kinds of MPs. Although the molecular weight of the MPs is different, they all have the conserved motifs, including a signal peptide, a serine/threonine (Ser/Thr)-rich region, a glycosylphosphatidylinositol (GPI) anchor attach site, and a functional domain [9]. These MPs contain abundant O-glycosylation sites in Ser/Thr rich regions [10,11,12]. Immunoelectron microscopy analysis showed that most MPs were located near the inner cell wall [13]. Although MPs’ content in the polysaccharide capsule is small, it plays an important physiological role in fungi. Teixeira et al. demonstrated that the mannoprotein MP84 was involved in the *C. neoformans*-epithelial lung cells adhesion through a *Cryptococcus*–epithelial lung cell interaction assay [14]. The frequent occurrence of cryptococcosis in T-cell deficient patients underscores the importance of the host cell response, and antigens that can recognize and cause cell responses have attracted much attention [15,16,17]. *C. neoformans* MPs are highly immunogenic and can stimulate T-cell responses [12,18]. The mannoproteins MP88 and MP98 can stimulate T-cell responses, and the MP84 and MP115 proteins can react strongly with sera from AIDS patients infected by *C. neoformans* [9,10,14,19], indicating that the MPs may be the main antigen of *C. neoformans* to stimulate the T-cell response. The immune mechanism of MPs may lie in its ability to target and bind the mannose receptors on the surface of antigen-presenting cells (APC), and it is speculated that this mechanism can induce not only a cellular immune response but also induce humoral response [20], which plays an important role in indicating the development of the cryptococcal vaccine.

Other mannoproteins also play an important role in virulence and/or cell wall integrity of fungal pathogens. Cadieux et al. showed that the *C. neoformans* mannoprotein Cig1 helps get iron from heme and further proved that the Cig1 and the ferroxidase Cfo1 jointly regulate the virulence of *C. neoformans* [21]. Reuwsaat et al. found that the predicted mannoprotein Krp1 is required for the cell wall structural integrity but is not necessary for fungal virulence *in vivo* in *C. gattii* [22]. Viudes et al. found that *Candida albicans* surface mannoprotein MP58 could elicit a strong antibody response during infection. The C-terminal antibody binding region of the MP58 protein is the antigenic epitopes of *C. albicans*, and monoclonal antibodies targeting this epitope can play a protective role in the serum treatment experiments in a mouse model of candidiasis [23]. In *Candida glabrata*, the mannoprotein Tir3 is necessary for sterol uptake [24]. In *Talaromyces* (*Penicillium*) *marneffei*, the mannoprotein Mp1p was proved to be an essential virulence factor, mediating virulence by enhancing *T. marneffei*’s survival inside macrophages [25,26].

The above results showed that although mannoprotein is not the main structural substance of fungal cells, it affects the growth, development, and virulence of fungal pathogens, and some mannoproteins can also act as specific antigens to stimulate the cellular immune response of the host and perform immune protection on the host.

In our previous work on identifying downstream targets of Fbp1, an F-box protein, we found a highly enriched protein Cmp1 (CNAG_06000) in *fbp1*Δ mutant background. The protein sequence analysis indicated that this protein has a high similarity with the mannoprotein MP88 of *C. neoformans* and *T. asahii*, and might be a mannoprotein. Since mannoproteins can develop protective immunity, we hypothesized that Cmp1 is a virulence factor that regulates the virulence of *C. neoformans*. In the present study, we knocked out and overexpressed the *CMP1*gene and examined its virulence role in a murine model to test our hypothesis. We demonstrated that the mannoprotein Cmp1 affects capsule production and regulates fungal virulence in *C. neoformans*.

## 2. Results

### 2.1. Identification of the Mannoprotein Cmp1 in C. neoformans

In our previous study, we used an iTRAQ-based proteomics approach to identify the downstream targets of Fbp1, a key protein of the E3 ubiquitin ligase. One of the candidate proteins, CNAG_06000, might be a substrate of Fbp1 because of the putative PEST (proline (P), glutamate (E), serine (S), and threonine (T)) domains in its protein sequences (Table 1, Figure 1). To investigate the function of cryptococcal CNAG_06000, we blasted the gene against the FungiDB database [27] and found that the CNAG_06000 gene is 1968 bp long and consists of 8 exons encoding a protein of 489 amino acids (Figure 1A, B). Analysis of the CNAG_06000 amino acid sequence showed that it contains one Ser-rich domain (Ser_RICH), one transmembrane domain (TR), five N-linked glycosylation sites (NGS), and one glycosyl-phosphatidylinositol (GPI)-anchor site (𝛚; Figure 1B). Smart blast analysis revealed that the CNAG_06000 protein shows 68% and 70% sequence similarities to the mannoprotein MP88 in *Trichosporon asahii* and *C. neoformans*, respectively (Figure 1C); we named the protein Cmp1 (*Cryptococcus* mannoprotein 1). The further protein sequences analysis indicates that the Cmp1 lacks signal peptide compared with the MP88s in *C. neoformans* and *T. asahii. Cryptococcus* MP88, as a specific antigen, stimulates the host’s cellular immune response and plays an immunoprotective role on the host [19]. Due to the importance of mannoproteins in the *C. neoformans*–host interaction, we decided to explore the function of Cmp1 in fungal development and virulence of *C. neoformans*.

### 2.2. CMP1 Expression Pattern Analyses

To analyze the *Cryptococcus CMP1* gene expression pattern, we cloned a 1.7 Kb native promoter of the *CMP1* gene first and fused it with mCherry (*P_CMP1_-mCherry)* to explore the temporal characteristics of the *CMP1* gene expression. The mCherry signals were detected in yeast cells, mating hyphae, basidium, and spores of P_CMP1_-mCherry strains (TBL209 and TBL175, see Table 2 for detailed information), indicating that the *CMP1* gene was expressed in all stages of *Cryptococcus* development (Figure 2A). Meanwhile, the expression of *CMP1* during mating was also analyzed by qRT-PCR. The mating mixtures between H99 (*MAT*⍺) and KN99 (*MAT***a**), the two wild-type strains, were harvested from V8 plates after designated incubation times. Compared with the 0-h time point’s gene expression level, our qRT-PCR data revealed that the *CMP1′*s expression was first upregulated and then downregulated during the process of mating, suggesting its possible role in the sexual reproduction of *C. neoformans* (Figure 2B). Since Cmp1 might be a mannoprotein, we then tested the expression of the *CMP1* gene under the capsule-inducing conditions, and the results showed that when *Cryptococcus* was cultured in YNB (yeast nitrogen base without amino acids) or DME (Dulbecco’s modified Eagle) medium, the expression of the *CMP1* gene was highly induced compared with that of *CMP1* in YPD (yeast extract peptone dextrose). However, when *Cryptococcus* was cultured in the MM (minimal medium) medium, the expression levels of the *CMP1* gene showed no difference from that of *Cryptococcus* cultured in the YPD medium (Figure 2C).

### 2.3. Cmp1 Localization in C. neoformans

To detect the subcellular localization of cryptococcal Cmp1, we constructed the *GFP-CMP1* fusion expression vector (pTBL92) and transformed it into the two *cmp1*Δ mutant strains (TBL106 and TBL137). Different from GFP-Nop1′s nuclear localization (TBL84) [32] (Figure 3A), the fusion GFP-Cmp1 protein was distributed unevenly in the cytoplasm of *Cryptococcus* yeast cells and mating hyphae (Figure 3A). Next, we also examined the cellular localization of GFP-Cmp1 fusion protein under various stress conditions including high-temperature, hyperosmosis, oxidative, or nitrosative stresses. Our data suggested that the subcellular localization of GFP-Cmp1 did not change under the above stress conditions (Figure 3B). Meanwhile, we also examined the cellular localization of GFP-Cmp1 under capsule induction conditions. Interestingly, we found that the GFP-Cmp1 fusion protein localized in the vacuole of cryptococcal yeast cells (Figure 3C). 

Since MM medium is a nutrient starvation medium used for capsule induction, it may affect the cellular localization of GFP-Cmp1. We then examined the cellular localization of the GFP-Cmp1 fusion protein in media such as YNB and DMEM. Our results showed that the GFP-Cmp1 fusion protein was localized in the cytoplasm of cryptococcal cells when the fluorescent strain was cultured in the YPD, YNB, or DME medium, while the GFP-Cmp1 fusion protein was localized in the vacuole when the fluorescent strain was cultured in the MM medium (Figure 3D). Our results indicate that the Cmp1 protein does have different subcellular localization under different culture conditions.

### 2.4. Cmp1 Regulates Capsule Formation

To identify the role of Cmp1 in *C. neoformans*, we created the *CMP1* gene disruption mutants *cmp1*Δ in the two wild-type strains backgrounds (Figure 4A–D). The *cmp1*Δ mutants complemented strain *cmp1*Δ*::CMP1* (TBL177 and TBL178, Figure 4E–G) and *CMP1* overexpressed strains *CMP1^OE^* (TBL186 and TBL187) were also generated. The complementation or overexpression of the *CMP1* gene was also verified by qRT-PCR (Figure 4G and Figure 5B). Since Cmp1 is a putative mannoprotein that may play a part in capsule formation, we first examined the role of Cmp1 in capsule formation, melanin production, and growth at 37 °C in vitro. The *cmp1*Δ mutants and *CMP1^OE^* strains generated normal melanin on Niger-seed agar plates and grew normally at 37 °C (Figure 5A and Figure 6) compared to that of the H99 strain. However, both the *cmp1*Δ mutants and *CMP1* overexpressed strains exhibit reduced capsule size when cultured in DME, MM, and RPMI (Roswell Park Memorial Institute 1460)-containing medium. Based on a Student’s *t*-test, the difference of the capsule size between the two strains and the H99 strain was significant, which indicated that Cmp1 is vital for the development of capsule in vitro (Figure 5C–E).

Then we investigated the *cmp1*Δ mutants and *CMP1*^OE^ strains grown in various stresses, including starvation stress (YNB), high-temperature stress (37 °C), cell wall stress (0.5% Congo red or 0.025% SDS), osmotic stress (1.5 M Sorbitol or 1.5 M NaCl), oxidative stress (2.5 mM H_2_O_2_), nitrosative stress (1 mM NaNO_2_, pH = 4.0), and drugs (Hyg and HU). Our results revealed that under the above stresses, there was no difference between the growth of *cmp1*Δ mutants and the *CMP1* overexpressed strains compared with the wild-type strains, indicating that Cmp1 did not participate in response to the above stress conditions (Figure 6).

### 2.5. Cmp1 is not Required for Sexual Reproduction

*C. neoformans* is a heterothallic basidiomycete having two mating types: *MAT***a** and *MAT*α. The haploid cells of two opposite mating types can fuse and undergo sexual reproduction, generating the dikaryotic mating hyphae and the haploid basidiospores. To confirm whether Cmp1 functions in cryptococcal sexual reproduction, we generated the *cmp1*Δ mutants and *CMP1* overexpressed strains in both H99 and KN99**a** strains. The mating filaments and basidiospores production was investigated in both *cmp1*Δ mutants bilateral mating (*cmp1*Δ × *cmp1*Δ) and unilateral mating (*cmp1*Δ × wild type). However, no noticeable phenotypic changes were observed in both *cmp1*Δ bilateral or unilateral-mating assays than the wild-type strains (Figure 7). Additionally, the *CMP1^OE^* strains generated normal mating filaments and basidiospores in bilateral mating (Figure 7). These data suggest that Cmp1 is not necessary in the sexual reproduction of *C. neoformans*.

### 2.6. Cmp1 is Required for Fungal Infection

Since Cmp1 regulates the formation of the capsule of *C. neoformans*, we speculated that Cmp1 might play a part in the fungal virulence of *C. neoformans.* To examine our prediction, we explored the virulence of *cmp1*Δ mutant and *CMP1^OE^* strain in a murine inhalation model of cryptococcosis. Ten female C57 BL/6 mice in each group were infected intranasally with 10^5^ cells of each cryptococcal strain, and the mice were examined twice a day. Consistent with our previous results, all the mice inoculated with the wild-type strain were terminated with a lethal infection between 18 and 25 dpi. By contrast, the *cmp1*Δ mutant’s virulence was attenuated significantly, and five of the *cmp1*Δ mutants-infected mice survived at the endpoint of the animal experiment (80 dpi; Figure 8A), which was remarkably different from that of wild-type control (*p* < 0.0001, log-rank (Mantel–Cox) test). The *cmp1*Δ mutant complemented strains developed a lethal infection in mice approximately 21–29 dpi, which exhibited no marked difference with the H99 strain (*p* > 0.9999, log-rank (Mantel–Cox) test), confirming that the attenuation of virulence in the *cmp1*Δ mutant was caused by the disruption of the *CMP1* gene (Figure 8A). Interestingly, the *CMP1^OE^* strains also showed significant virulence defect compared with the wild-type H99 strain, (*p* < 0.0001, log-rank (Mantel–Cox) test; Figure 8A).

To find out why was the virulence of the *cmp1*Δ mutant and the *CMP1*^OE^ strain attenuated, we examined the fungal load of the infected mice when the infection experiment was terminated. The brains, lungs, and spleens of five mice infected with each *cryptococcal* strain were dissected, and the fungal load of these organs was assessed as yeast CFU per gram fresh organs. Our results indicated that 10^7^, 10^9^, and 10^7^ CFU were recovered from the brains, lungs, and spleens at the endpoint of the infection experiments, respectively when the mice inoculated with the wild-type H99 strain (Figure 8B). Organs of the mice infected by the *cmp1*Δ mutants were also dissected to calculate the fungal burdens. From the brains, lungs, and spleens of the mice sacrificed before 80 dpi, 10^7^, 10^8^, and 10^7^ CFU were recovered. In contrast, only about 10^2^ and 10^3^ CFU were isolated from the brain and lung, and no cryptococcal cells were found from the spleens of the mice sacrificed at 80 dpi. Organs of the mice infected by the *CMP1*^OE^ strains were also dissected to calculate the fungal burden, and 10^7^, 10^9^, and 10^7^ CFU were isolated from the brains, lungs, and spleens, respectively (Figure 8B).

Additionally, hematoxylin–eosin stained slides were prepared to visualize the development of fungal lesions in the organs of mice infected by *Cryptococcus*. As shown in Figure 8C, infection by both the wild-type strain and the *cmp1*Δ::*CMP1* complemented strain resulted in severe damage in mouse brains at the infection experiment‘s endpoint. However, it took more than 50 days for the *cmp1*Δ mutant to cause similar damage, showing that the *cmp1*Δ mutant’s virulence was significantly attenuated. By contrast, the *cmp1*Δ mutants can only cause very little damage to the infected brain of the mice sacrificed at 80 dpi, indicating that the *cmp1*Δ mutants’ virulence was significantly attenuated.

Meanwhile, in the lungs infected by the H99 or the *cmp1*Δ::*CMP1* complemented strain at 23 dpi, and *cmp1*Δ mutants at 50 dpi, large numbers of cryptococcal cells with thickened capsules can be visualized to cause severe damage to the organs. However, no detectable damage was visualized in the lungs of the *cmp1*Δ-infected mice that survived even at 80 dpi (Figure 8C). Similar results were also observed in the spleens infected by the above cryptococcal strains (Figure 8C). These data suggest that the Cmp1 is indispensable for the development of cryptococcosis in a mouse model.

To find out why the *cmp1*Δ mutants are virulence defective, we examined the fungal loads in infected mice’s organs at 7, 14, and 21 dpi. CFU counts results show that the *cmp1*Δ mutant could still cause organ infection in the infected mice, but the CFUs recovered from organs of the mice infected by the *cmp1*Δ mutant were significantly lower than that of the wild-type H99 strain (Figure 9A–C). Histopathologic findings also indicated that the *cmp1*Δ mutants could still cause lung infections, but compared with the wild type or the *cmp1*Δ::*CMP1* complemented strains infected mice, the cell density was smaller and lesion development happened later and was less extensive (Figure 9D, and data not shown).

### 2.7. Cmp1 is Important for Proliferation Inside Macrophage and Survival in the Host Complement System

Our virulence study showed that the *cmp1*Δ mutant’s virulence was significantly attenuated in a mouse systematic inhalation model. Since macrophages and monocytes are the first host cells that *C. neoformans* encountered after it gets into the host lung, we hypothesized that the *cmp1*Δ mutant might have proliferation defects in host macrophages, and it is difficult for extracellular cryptococcal cells to survive in the hostile environment inside the host. To test our hypotheses, we first carried out a *Cryptococcus*–macrophage interaction assay using the RAW246.7 murine macrophage cells. After two hours of coincubation of the cryptococcal cells and macrophages, the yeast CFU number recovered from the *cmp1*Δ mutant-coincubated macrophages was comparable to that recovered from the macrophages coincubated with wild-type strains, indicating a similar phagocytosis level between the *cmp1*Δ mutant and the wild-type strains (Figure 10A).

However, after 24 h of incubation, the yeast CFU recovered from the *cmp1*Δ mutant-interacting macrophages was significantly fewer than that of the wild-type-interacting macrophages (*p* < 0.001, Figure 10A). These results suggested that the *cmp1*Δ mutant proliferates slower than the wild-type strain once it is engulfed by macrophages, which could be one reason why the *cmp1*Δ mutants have significant virulence attenuation in the mouse systemic infection model.

Meanwhile, we also tested the *Cryptococcus* cells’ viability after incubation with mouse serum for 1, 2, 3, and 4 h to verify whether components of the host complement system damage *C. neoformans*. Our results showed that the survival rate of *cmp1*Δ mutants was significantly lower than that of the wild-type strains after 4 h incubation of *Cryptococcus* cells with the mouse serum (*p* < 0.001, Figure 10B), which also indicated that components of the host complement system did have more severe damage on *cmp1*Δ mutants.

The above results indicated that the disruption of the *CMP1* gene might affect *Cryptococcus’* virulence by affecting the proliferation inside macrophages or affecting the viability of *Cryptococcus* in the host complement system.

## 3. Discussion

Despite its low content, mannoprotein is a crucial component of the *Cryptococcus* polysaccharide capsule, stimulating the host’s immune protection. In this study, we identified and functionally analyzed a predicted mannoprotein Cmp1, and our results showed that the Cmp1 is a predicted mannoprotein and is required for fungal virulence in *C. neoformans*.

In the literature, at least four mannoproteins (MP84, MP88, MP98, and MP115) in *C. neoformans* have been suggested to play a part in stimulating host T-cell responses [10,14,19,33]. The fifth mannoprotein, Cig1, has been proved to help increase the virulence of *C. neoformans* by supporting iron acquisition from heme [21]. Among these five mannoproteins, only Cig1 was tested and shown to have virulence properties in mouse models. In our previous work identifying the downstream targets of the F-box protein Fbp1, we found that Cmp1 might be a downstream substrate of Fbp1 due to its high abundance in *fbp1*Δ mutant background (Table 1). However, when we did a sequence analysis of Cmp1, we found that Cmp1 did not have the signal peptide sequence typical of mannose glycoproteins (Figure 1C). Unlike other mannoproteins, e.g., MP98, located near the cell wall [9], subcellular localization indicates that Cmp1 is located in the cytoplasm. More interestingly, however, when we examined the subcellular localization of Cmp1 under nutrient or starvation conditions, we found that the localization of Cmp1 differs depending on the culture conditions. When cultured in nutrient media such as YPD, YNB, and DMEM, the Cmp1 proteins localized in the cytoplasm of the *Cryptococcus* cells. However, the Cmp1 proteins is localized in the vacuoles when cultured in the capsule-inducing medium, e.g., MM, which might be because the capsule-inducing medium is a nutrient-deficient medium, and *Cryptococcus* cells cannot get enough nutrients from it and has to degrade its own substances, which causes the GFP-cmp1 protein to accumulate in the vacuole. Our results suggest that Cmp1 might be a mannoprotein that does not contain a signal peptide and cannot be secreted outside the cell.

In the opportunistic dimorphic fungus *Talaromyces marneffei*, Mp1p, and its 13 Mp1p homologs (MpLp1-13) were identified in its genome [26]. Among the 13 Mp1p homologs, the MpLp7 has no signal peptide sequence, while the remaining 12 Mp1p homologues contain putative signal peptides, varying amounts of putative O- and N-glycosylation sites, or glycosylphosphatidylinositol (GPI) anchors [26]. The presence of mannoprotein without a signal peptide in *T. marneffei* is also suggested that the *Cryptococcus* Cmp1 might be a mannoprotein without a signal peptide. Meanwhile, our results also showed that the *CMP1* gene knockout or overexpression could affect the capsule’s formation, one of the three virulence factors. However, since Cmp1 is an intracellular localized protein and is located in vacuoles under the conditions that induce capsule formation, how Cmp1 affects *Cryptococcus* capsule formation remains unknown. Thus, further research is needed to determine how Cmp1 affects cryptococcal capsule formation.

The molecular mechanism of Cmp1 regulating virulence is unclear and remains to be determined. Fungal virulence assay in this study showed that both disruption and overexpression of *CMP1* resulted in significant virulence attenuation in *C. neoformans* (Figure 8). Progression of *cmp1*Δ mutant infection in vivo at 7, 14, and 21 dpi indicated that while the *cmp1*Δ mutant could still cause infection in infected organs, the cell density was smaller and lesion developed later and was less extensive compared with the H99 infected mice (Figure 9). However, these data cannot explain the molecular mechanism by which Cmp1 controls the virulence of *C. neoformans*. Disruption or overexpression of *CMP1* affects capsule formation in *C. neoformans*, but whether this directly affects *Cryptococcus* virulence is still unknown.

Macrophages are the key defensive cells against *Cryptococcus* infection. In our study, the *cmp1*Δ mutant’s virulence was significantly attenuated in a mouse systematic infection model. Since macrophages or monocytes are the first host cells that *C. neoformans* encountered after it gets into the host lung, we hypothesized that the *cmp1*Δ mutant might have proliferation defects in host macrophages. Our *Cryptococcus*-macrophage interaction assay did show that the *cmp1*Δ mutant proliferates slower than the wild-type strain once it is engulfed by macrophages (Figure 10A). Besides, the *Cryptococcus* cells’ viability assay in our study showed that the survival rate of *cmp1*Δ mutants was significantly lower than that of the wild-type strains after incubation with mouse serum (Figure 10B). Both of the above two experimental results showed that disruption of the *CMP1* gene significantly affect the proliferation or survival of *Cryptococcus* cells in the host, which could be one reason why the *cmp1*Δ mutants have a significant virulence attenuation in the mouse systemic infection model.

Fungal pathogens, such as *C. neoformans*, *A. fumigatus, C. albicans,* and *T. marneffei* are highly virulent fungi, causing high fatalities in HIV-positive patients, transplant recipients, and patients receiving corticosteroid therapy [34,35]. However, there are relatively few antifungal drugs currently, and the latest antifungal drugs have shown resistance in these pathogens. The fungal cell wall made up of ⍺ and β glucans, chitin, chitosan, and mannoproteins, is vital for fungal growth and development and is also one of the most significant drug development targets [36,37]. The results of previous studies showed that mannoproteins play a vital role in fungal virulence or cell wall integrity, such as Cig1 in *C. neoformans*, Krp1 in *C. gattii*, and Mp1p in *T. marneffei* [21,22,26]. Given mannoproteins’ role in fungal virulence or cell wall integrity in fungal pathogens, it might be one of the potential targets for antifungal drug development.

## 4. Materials and Methods

### 4.1. Ethics Statement

The animal experiments conducted at the Southwest University were in complete accordance with the Ministry of Science and Technology of China’s “Guidelines for the Ethical Care of Laboratory Animals (No. 398, 2006)”. Southwest University’s Animal Ethics Committee approved all vertebrate experiments.

### 4.2. Strains and Growth Conditions

*Cryptococcus* strains used in the present study are shown in Table 2. The wild-type strains of *Cryptococcus* H99 and KN99**a** were routinely preserved in our laboratory. All other cryptococcal strains were derived from wild-type strains. Cryptococcal strains are grown on YPD medium at 30 °C or indicated temperatures. MS or V8 medium were used for induction of mating and prepared as described previously [38]. DME medium, minimal medium (MM), and RPMI-contaning medium were used for the induction of *Cryptococcus* capsule formation [39,40,41]. All other media used in this study were prepared as previously described [39].

### 4.3. CMP1 Expression Pattern Assay

To investigate the temporal expression *CMP1*, we amplified a 1738 bp *BamH*I/*BamH*I promoter fragment of the *CMP1* gene using the wild-type genomic DNA as templates with primers TL481/482 and inserted into the plasmid pTBL3 [31] to construct the *CMP1* promoter and mCherry fusion vector pTBL82 (see Table 3 for plasmid information). To explore the cellular location of cryptococcal Cmp1 protein, we amplified the coding region of the *CMP1* from the wild-type genomic DNA using primers TL554/555 and inserted into plasmid pCN19 to construct the *GFP*-*CMP1* fusion expression vector, as previously described [31]. The resulting plasmids, pTBL82, and pTBL92 were linearized with *Sca*I and *Xmn*I, respectively, and biolistically transformed into the wild-type strains, H99 and KN99**a**, as described previously [31,42]. Stable transformants were further confirmed on the YPD containing 100 mg/liter nourseothricin sulfate. The transformants’ fluorescence was visualized by confocal microscopy (Olympus, FV1200).

Meanwhile, to detect the *CMP1* gene expression during cryptococcal mating, we used qRT-PCR to detect the *CMP1* gene’s expression at mRNA levels. Mating cultures preparation, total RNAs isolation and purification, and cDNAs synthesis were carried out as described previously [31]. Gene expression of *CMP1* was analyzed by the comparative C_T_ approach using the *GAPDH* gene as an internal control, as described previously [31].

### 4.4. Generation of CMP1 Deletion, Complementation, and Overexpression Strains

The *CMP1* gene was disrupted in both wild-type strains using a split marker strategy, as described previously [31,43]. Primers TL235/TL236 and TL237/TL238 were used to amplify the 5′ and 3′ regions of the *CMP1* gene, respectively, using H99 genomic DNA as templates (for primer sequences, see Table 2). The M13 primers (TL17 and TL18) were used to amplify the NEO dominant selectable marker using the plasmid pJAF1 as templates [44]. The 5′ or 3′-region of the NEO-split marker containing gene disruption cassettes was amplified by primers TL235/TL20 or TL19/238, respectively, using the mixture of *NEO* marker and *CMP1* 5′ region or 3′ region as templates, respectively. The overlap PCR fragments were purified, combined, and then biolistically transformed into the H99 or KN99**a** strains, as described previously [31]. Stable transformants were further confirmed on the YPD medium containing 200 mg/liter G418. The *cmp1*Δ mutants were further confirmed by diagnostic PCR (for primer sequences, see Table 2) and Southern blotting analysis.

To obtain the *cmp1*Δ complemented strain, we amplified a 4.1 Kb genomic DNA fragment containing the *CMP1′*s upstream promoter region, open reading frame (ORF), and downstream terminator region using primers TL562/TL563 and cloned it into the plasmid pTBL1 to generate pTBL94 by infusion cloning. The resulting plasmid pTBL94, which contains the NAT selective marker, was digested by *Sca*I and biolistically transformed into *cmp1*Δ mutants to generate the complemented strains.

To overexpress the *Cryptococcus CMP1* gene, we linearized the plasmid pTBL92, which contains the *GFP-CMP1* fusion construct under the histone H3 promoter’s control, with *Xmn*I and biolistically transformed it into the *cmp1*Δ mutants. The overexpression of *CMP1* was confirmed by qRT-PCR and fluorescence observation.

### 4.5. Assays of Melanin and Capsule Production, and Mating

Melanin formation was tested by dripping 50 microliters of each PBS-washed *Cryptococcus* overnight culture onto the Niger-seed agar plates. The Niger-seed plates were then placed at 30 °C or 37 °C and incubated for two days, and the fungal colonies pigmentation was evaluated and photographed.

To investigate the capsule formation, we inoculated 10^6^ cells cultured in YPD medium overnight into MM medium and cultured at 30 °C for 72 h. The capsule’s size was assessed, as previously described [39]. For matings, cell suspensions of cryptococcal mating partners (⍺ or **a**) were combined and cultured on V8 or MS medium in darkness at 25 °C. Formation of mating hyphae and chains of basidiospores were observed and recorded microscopically after incubation for two weeks.

### 4.6. Virulence Studies

Each strain’s overnight culture was washed twice with PBS buffer and diluted to 2 × 10^6^ cells/mL. Eight-week-old C57 BL/6 mice (female, *n* = 10/group) were intranasally inoculated with 10^5^ CFU of each *Cryptococcus* strain, as described previously [45]. During the animal experiments, mice that were appeared to be in pain or moribund were euthanized by carbon dioxide inhalation. The log-rank test was used to calculate survival statistics between paired groups, and the nonparametric Mann–Whitney test with PRISM 7.0 (GraphPad Software, San Diego, CA, USA) was used for fungal load’s statistical analyses.

### 4.7. Fungal Loads and Histopathology of Infected Organs

According to an animal protocol approved by Southwest University, the infected mice were euthanized at the endpoint of the animal experiment. For mice inoculated with the *cmp1*Δ mutant, the animal experiment was stopped at 80 dpi. To compare the host inflammatory responses and fungal loads, we dissected the organs from the animals infected by each *Cryptococcus* strain at the animal experiments’ endpoint. Infected organs were dissected and homogenized in PBS buffer with a tissue homogenizer. The homogenates were diluted, and 100 microliters of each diluent were dispersed on the YPD medium containing ampicillin and chloramphenicol. The CFUs of the infected organs were calculated after incubation at 30 °C for three days. The infected organs were also sent to Servicebio (Servicebio, Wuhan, China) for section preparation after fixation in 10% formalin solution. Hematoxylin and eosin (H&E)-stained tissue slides were observed under the light microscope.

### 4.8. Cryptococcus-Macrophage Interaction and Serum Treatment Assay

The *Cryptococcus*–macrophage interaction assay was carried out as described previously [39,46]. RAW246.7 murine macrophage cells were used in this study. After washing with PBS and opsonization with 20% mouse complement, a total of 2 × 10^5^ cryptococcal cells of the overnight cultures of each *Cryptococcus* strain were added to each well containing RAW246.7 macrophage cells. After 2 h of coincubation at 37 °C, some of the cells were washed and fixed for phagocytosis assays as described previously [39]. Meanwhile, fresh DMEM was used to wash away the nonadherent extracellular cryptococcal cells, and the cultures were incubated for 0, 2, or 22 more hours. At various time points, distilled water was added to each well after removing the DMEM to lyse macrophage cells. The lysate was dispersed on YPD plates and yeast CFUs were counted to determine the phagocytosis rate and intracellular proliferation.

Serum treatment and *Cryptococcus* cell viability assay was performed as described previously [47]. At the indicated time points, aliquots were taken out and plated to YPD medium after serial dilution to determine cell viability.

## Figures and Tables

**Figure 1 pathogens-09-00881-f001:**
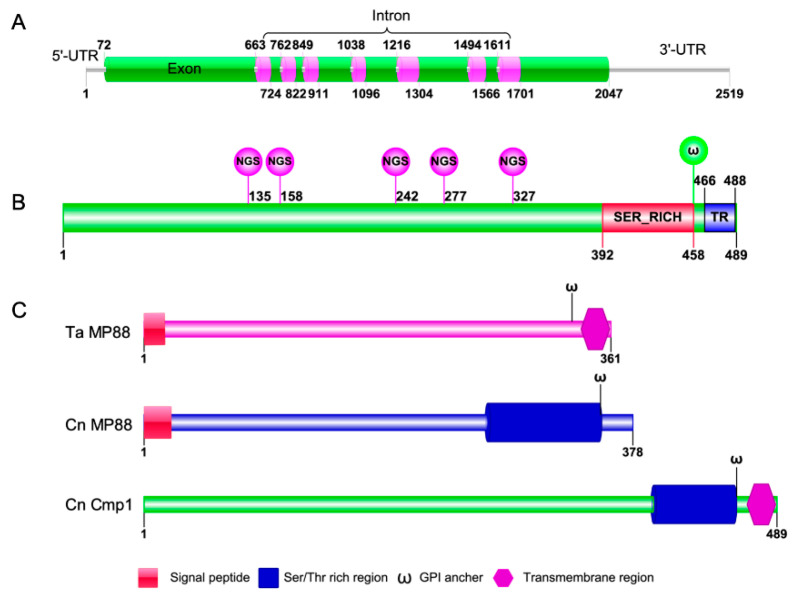
Sequence analysis of the *Cryptococcus* mannoprotein Cmp1. (**A**) The sequence structure of the *Cryptococcus* mannoprotein encoding gene *CMP1*. (**B**) Functional domains or sites of the *Cryptococcus* mannoprotein Cmp1. NGS: Glycosylation site; SER_RICH: ser/thr rich region; 𝛚: GPI anchor; TR: Transmembrane region. (**C**) Comparison of protein structures between *C. neoformans* Cmp1 (Cn Cmp1), *C. neoformans* MP88 (Cn MP88), and *T. asahil* MP88 (Ta MP88).

**Figure 2 pathogens-09-00881-f002:**
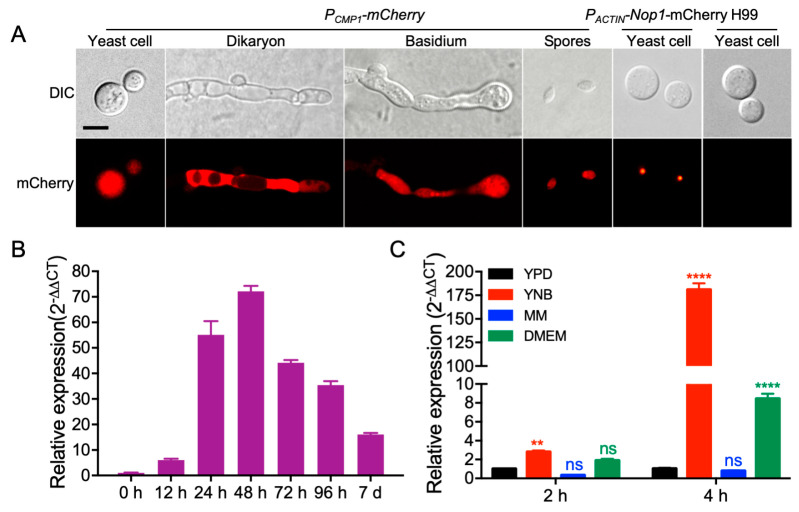
The expression pattern of the cryptococcal *CMP1* gene. (**A**) The expression of *P_CMP1_-mCherry* during the mating was visualized by confocal microscopy (Olympus, FV1200, Tokyo, Japan). Representative images of yeast cells, mating hyphae, basidia, and basidiospores of both bright-field and fluorescence are shown. Scale bar, 5 μm. (**B**) Quantitative real-time PCR (qRT-PCR) was used to analyze the expression of *CMP1* during mating. Mating mixtures of the wild-type strains were harvested from the V8 plates after incubation for indicated times and used for RNA extraction, and cDNA was synthesized. The comparative cycle threshold (CT) approach was used to quantify gene expression, and the *GAPDH* gene was used as an internal reference. The experiment was repeated three times. (**C**) The expression of the *CMP1* gene under nutrient or starvation condition. Overnight cultures of the H99 strain were washed and transferred to new YPD, YNB, MM, and DME medium and culture for 2 and 4 h. Relative qRT-PCR was used to detect the expression of *CMP1* under nutrient or starvation conditions, as described in (B). ns: not significant; **, *p* < 0.01; ****, *p* < 0.0001.

**Figure 3 pathogens-09-00881-f003:**
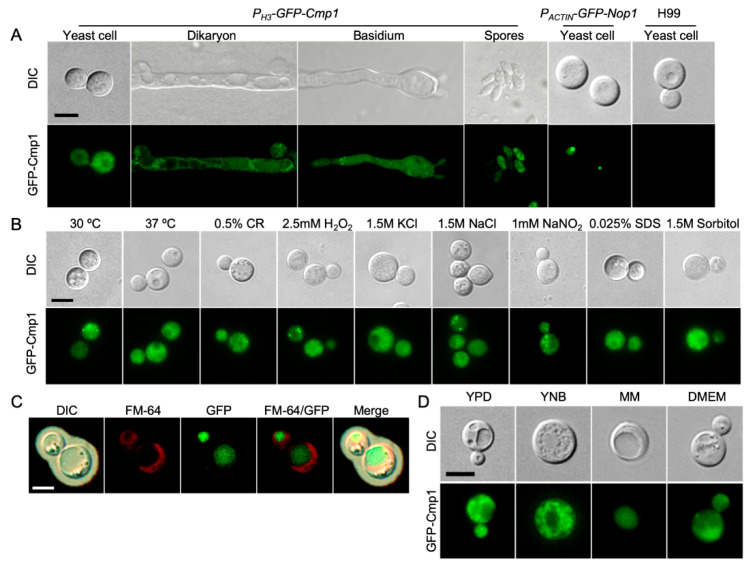
Subcellular localization of *Cryptococcus* Cmp1. (**A**) The yeast cells, dikaryon, basidium, and basidiospores of the strain expressing GFP-Cmp1 were observed by confocal microscopy (Olympus, FV1200). The fusion protein of GFP and *Cryptococcus* Cmp1 (GFP-Cmp1) is localized in the cytoplasm of *C. neoformans*. Yeast cells of the nuclear-localized GFP-Nop1 strain and the wild-type H99 strain were positive and negative controls. Scale bar, 5 μm. (**B**) GFP-Cmp1′s localization in yeast cells under different stresses. CR: Congo red. Scale bar, 5 μm. (**C**) GFP-Cmp1′s localization in yeast cells after cultured in MM medium for three days. (**D**) GFP-Cmp1′s localization in yeast cells after cultured in capsule-inducing media for three days. Scale bar, 5 μm.

**Figure 4 pathogens-09-00881-f004:**
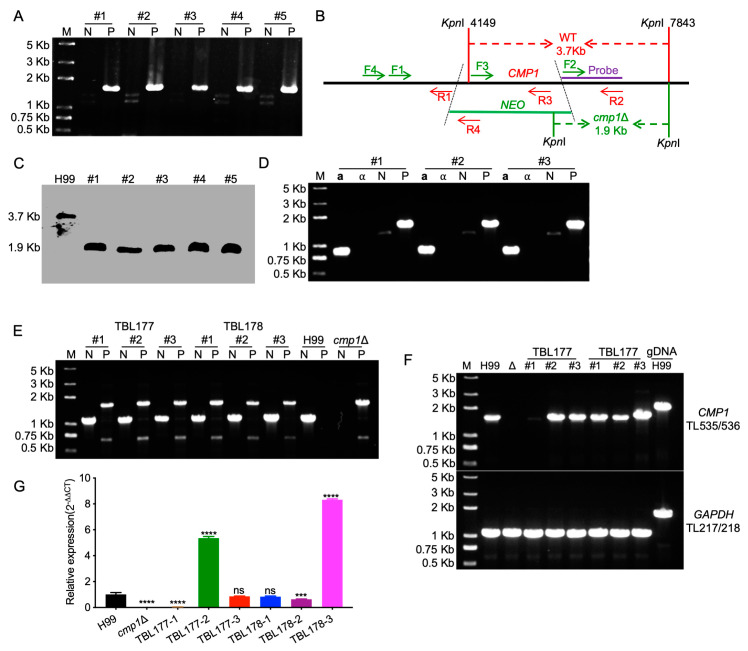
Generation and confirmation of *CMP1* gene knockout, complementation, and overexpression strains. (**A**) The diagnostic PCR confirmation of the five transformants with G418 resistance. P: primers for positive amplification, TL241/TL59 (F4/R4 in 3B); N: primers for negative amplification, TL239/TL240 (F3/R3 in 3B). (**B**) Restriction enzyme used in the digestion of the genomic DNAs in Southern blotting. The probe was synthesized using the TL237/TL238 (F2/R2) PCR products as templates. The *cmp1*Δ mutants will produce a 1.9 Kb band while the wild-type strain H99 generates a 3.7 Kb band. (**C**) Southern blotting analysis of the *CMP1* knockout mutants. An equal amount of genomic DNA (50 µg) was digested with *Kpn*I, fractionated, and hybridized with a *CMP1* downstream flanking sequence-specific probe, as shown in Figure 3B. As anticipated, a 1.9 and 3.7 Kb bands were generated in the *cmp1*Δ mutants and the wild-type strain H99, respectively. (**D**) Verification of the **a** mating-type *cmp1*Δ mutants. **a**: **a** mating type-specific primers, TL69/70; ⍺: ⍺ mating type-specific primers, TL67/68; N: primers for negative amplification, TL239/TL240 (F3/R3 in 3B); P: primers for positive amplification, TL241/TL59 (F4/R4 in 3B). (**E**–**G**) Generation and verification of *CMP1* complementation strains. The genome DNA (**E**) and cDNA (**F**) of the complementation transformants were used as templates for PCR amplification to verify the *CMP1* gene’s complementation. N: primers for negative amplification, TL239/TL240 (F3/R3 in 4B); P: primers for positive amplification, TL241/TL59 (F4/R4 in 4B); the expression of *CMP1* of the complementation transformants was measured by relative qRT-PCR. ns: not significant; ***, *p* < 0.001; ****, *p* < 0.0001.

**Figure 5 pathogens-09-00881-f005:**
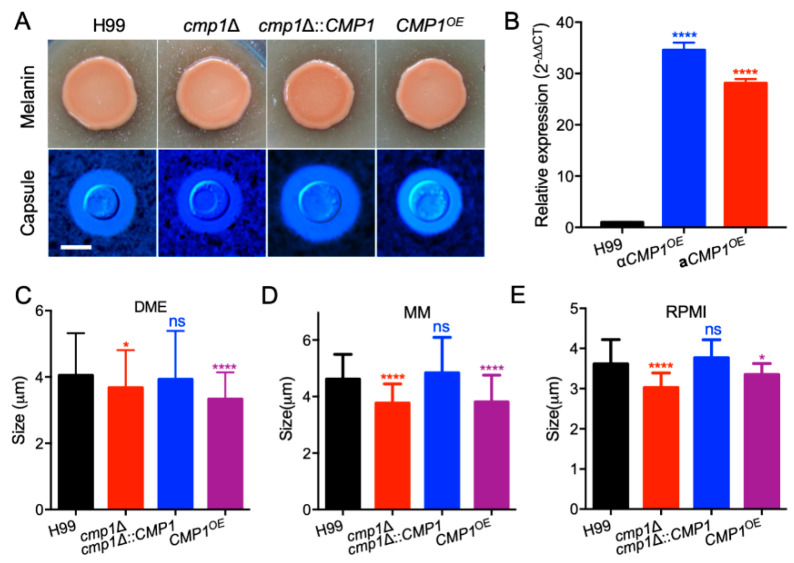
Cmp1 is necessary for *Cryptococcus* capsule formation. (**A**) Melanin formation (top) and capsule production (bottom) were carried out on Niger seed medium and DME medium, respectively. The melanin levels generated by the *Cryptococcus* strains were photographed after incubation at 37 °C for two days. The *Cryptococcus* cells were cultured in the DME medium at 37 °C for three days, and the formation of the capsule was observed by India ink staining. Bars, 5 μm. (**B**) The overexpression of the *CMP1* gene was evaluated by the relative qRT-PCR analysis. To compare the differences in gene expression, we arbitrarily set the expression levels of the *CMP1* gene in H99 strain grown on YPD medium as 1. ****, *p* < 0.0001. (**C**–**E**) Statistical analysis of cryptococcal capsule production in DME, MM, and RPMI-containing medium. The capsule size from at least 100 cells was measured, and the data shown are the average with standard deviation from three repeats. ns: not significant; *, *p* < 0.05; ****, *p* < 0.0001.

**Figure 6 pathogens-09-00881-f006:**
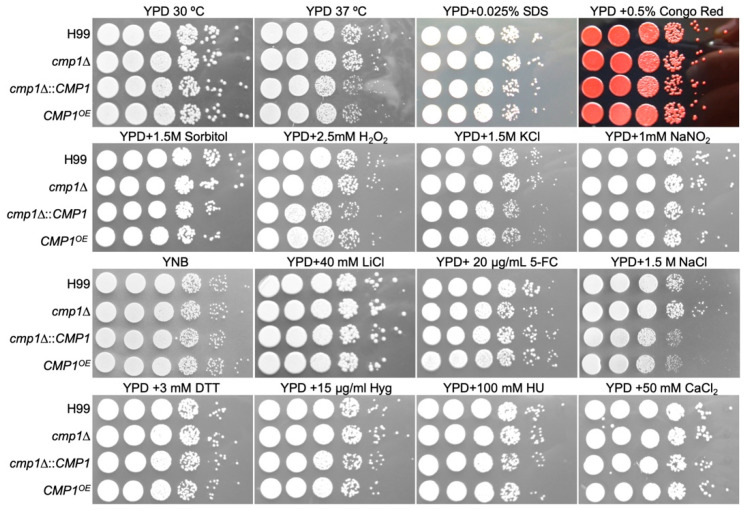
Growth of *cmp1*Δ mutants under different stress conditions. Each strain’s overnight culture was first diluted to an OD600 value of 2.0, and then ten-fold serial dilutions were prepared with ddH_2_O. Of each dilution 5 µL were dropped on YPD plates with different stresses and then incubated at 30 °C for 2 or 3 days. The culture conditions are shown at the top, and the *Cryptococcus* strains at the left. 5-FC: 5-fluorocytosine; DTT: dithiothreitol; HU: hydroxyurea.

**Figure 7 pathogens-09-00881-f007:**
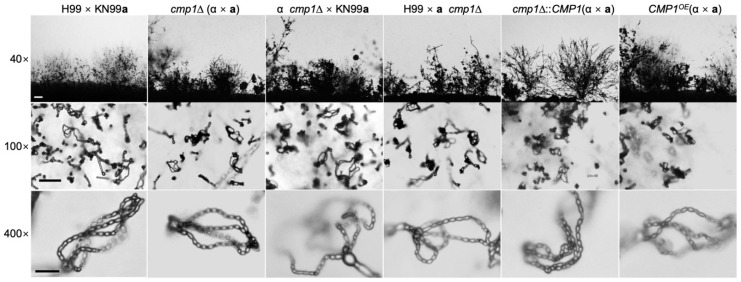
Deletion or overexpression of *CMP1* shows no effect on filamentation and sporulation during mating. Bilateral and unilateral mating assays were carried out on MS medium with the wild-type, *cmp1*Δ mutants, *cmp1*Δ complemented strains, and *CMP1* overexpression strains. After two weeks of incubation at 25 °C in the dark, images of the mating structures were captured at ×40 magnification (top, bar = 100 µm), ×100 magnification (middle, bar = 50 µm), and ×400 magnification (bottom, bar = 10 µm).

**Figure 8 pathogens-09-00881-f008:**
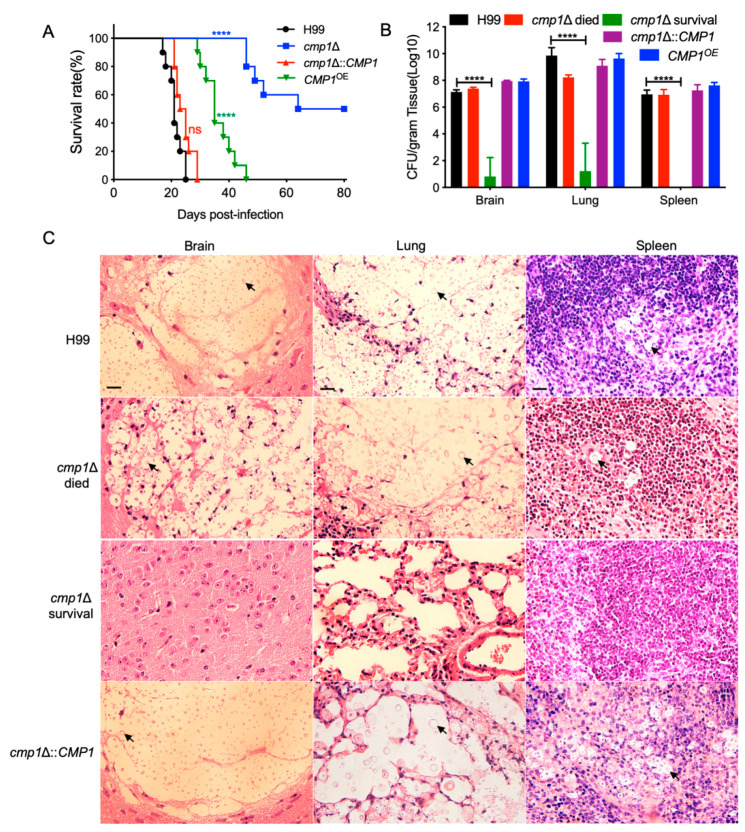
Cmp1 is required for full virulence of *C. neoformans*. Ten C57 BL/6 mice in each group were infected intranasally with 10^5^ cells of the wild-type H99, the *cmp1*Δ mutant, the *cmp1*Δ::*CMP1* complemented strain, and the *CMP1*^OE^ strain. (**A**) Survival rates of C57 BL/6 mice after infection. Both the *cmp1*Δ mutant and the *CMP1*^OE^ strain are less virulent than the wild-type H99 strain. ns: not significant; ****, *p* ≤ 0.0001 (determined by the log rank (Mantel–Cox) test). (**B**) Fungal load (CFU) in brains, lungs, and spleens. The data shown are the mean ± SD for the values of five mice. ****, *p* ≤ 0.0001 (determined by Mann–Whitney test). (**C**) Hematoxylin–eosin stained slides from the organs’ cross-sections at the end of the infection experiment were prepared and observed under a light microscope. Arrows indicate the cryptococcal cells. Scale bar, 20 μm.

**Figure 9 pathogens-09-00881-f009:**
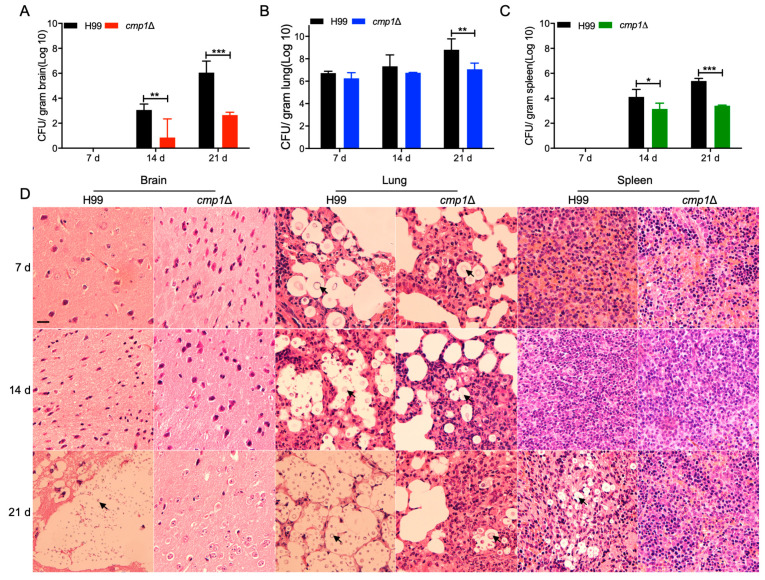
Progression of *cmp1*Δ mutant infection *in vivo*. Organs from mice inoculated with H99 or the *cmp1*Δ mutant were dissected at 7, 14, and 21 dpi. CFUs were calculated in the homogenates of the brains (**A**), lungs (**B**), and spleens (**C**). The data shown are the mean ± SD for values from five mice. *, *p* ≤ 0.05; **, *p* ≤ 0.01; ***, *p* ≤ 0.001 (determined by the Mann–Whitney test). (**D**) Hematoxylin–eosin stained slides from the cross-sections of the brains, lungs, and spleens were also prepared and observed by light microscopy. Arrows indicate the cryptococci. Bar, 20 μm.

**Figure 10 pathogens-09-00881-f010:**
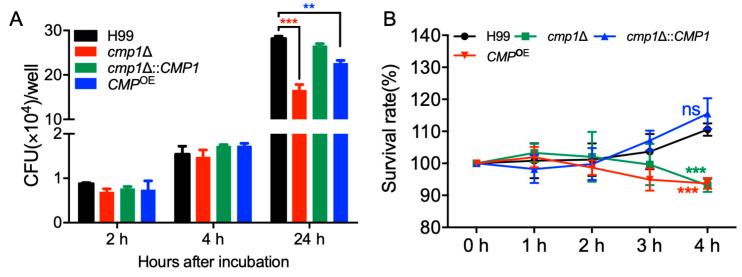
Cmp1 is important for proliferation inside macrophage and survival in the host complement system. (**A**) *Cryptococcus*’ proliferation inside macrophage was carried out using the RAW246.7 murine macrophage cells. PBS-washed *Cryptococcus* strains (H99, *cmp1*Δ mutant, *cmp1*Δ::*CMP1* and *CMP1*^OE^) were coincubated with activated macrophages for 2 h. The cells were incubated for an additional 0, 2, or 22 h after removing the nonadherent extracellular *Cryptococcus* cells by washing with fresh DMEM and then lysed by H_2_O for 30 min. The lysate was dispersed on the YPD plates, and the intracellular proliferation and macrophage killing were determined by CFU counting. **, *p* < 0.01; ***, *p* < 0.001. (**B**) Overnight cultures of the same strains as in (**A**) were washed twice with PBS buffer and diluted to a final concentration of 1 × 10^7^ CFU/mL. Fifty microliters of each strain’s cell suspensions were added to 450 microliters of mouse serum and incubated at 37 °C for indicated times. The precooled PBS buffer was added to stop the reaction, and the cells were washed twice with PBS buffer. One hundred microliters of the dilute (10^3^ dilution) was spread on YPD plates, and the number of CFU was used to measure the *Cryptococcus* cell viability. ns: not significant; ***, *p* < 0.001.

**Table 1 pathogens-09-00881-t001:** High-abundance proteins identified in the *fbp1*Δ mutants.

Accession	Description	Average *fbp1*Δ/H99	PEST Domain
CNAG_00626	Uncharacterized protein	3.02656981	1
CNAG_06000	Glycoprotein	1.87701854	1
CNAG_06195	Uncharacterized protein	1.59340506	0
CNAG_05395	Rab guanyl-nucleotide exchange factor	1.5884178	3
CNAG_00700	Purine nucleotide biosynthesis-related protein	1.54435399	0
CNAG_02455	Choline transporter	1.46895293	0
CNAG_06871	Uncharacterized protein	1.45686983	1
CNAG_02138	DNA replication ATP-dependent helicase Dna2	1.44213261	2
CNAG_01536	Nonmuscle myosin heavy chain b	1.3779496	1
CNAG_04056	Rhomboid-like protein	1.37338159	0
CNAG_04669	Mitochondrial matrix protein import protein	1.36855445	0
CNAG_03099	Chitin synthase 1	1.36597408	1
CNAG_01193	Uncharacterized protein	1.36434725	1
CNAG_05817	GDP-mannose transporter 1	1.35714493	0
CNAG_05967	Uncharacterized protein	1.35536384	1
CNAG_04327	SCP160 protein	1.35312155	1
CNAG_00634	Uncharacterized protein	1.3519146	1
CNAG_05173	DNA-3-methyladenine glycosylase II	1.34218534	0
CNAG_03281	Glycine-rich RNA binding protein, variant 2	1.33612571	0

**Table 2 pathogens-09-00881-t002:** Strains and plasmids used in this study.

Strains/Plasmids	Genotypes/Properties	Sources/References
*C. neoformans*		
H99	*MAT*α	Perfect et al., 1993 [28]
KN99a	*MAT*a	Nielsen et al., 2003 [29]
TBL106	*MAT*α *cmp1*Δ::*NEO*	In this study
TBL137	*MAT*a *cmp1*Δ::*NEO*	In this study
TBL175	*MAT*a *P_CMP1_-mCherry*::*NAT*	In this study
TBL177	*MAT*α *cmp1*Δ::*NEO CMP1*::*NAT*	In this study
TBL178	*MAT*a *cmp1*Δ::*NEO CMP1*::*NAT*	In this study
TBL186	*MAT*α *cmp1*Δ::*NEO P_H3_-GFP-CMP1*::*NAT*	In this study
TBL187	*MAT*a *cmp1*Δ::*NEO P_H3-_GFP-CMP1*::*NAT*	In this study
TBL209	*MAT*α *P_CMP1_-mCherry*::*NAT*	In this study
Plasmids		
pCN19	Amp^r^ Vector carrying *GFP* under the control of histone H3 promoter	Price et al., 2008 [30]
pTBL1	Amp^r^ Vector carrying *NAT* marker	Fan et al., 2019 [31]
pTBL3	Amp^r^ Vector carrying *mCherry-GPD1* terminator	Fan et al., 2019 [31]
pTBL82	Amp^r^ Vector carrying *P_CMP1_-mCherry-NAT* for temporal expression assay	In this study
pTBL92	Amp^r^ Vector carrying *P_H3_-GFP-CMP1* gene fusion for Cmp1 localization	In this study
pTBL94	Amp^r^ Vector carrying *P_CMP1_-CMP1-NAT* for *CMP1* complementation	In this study

**Table 3 pathogens-09-00881-t003:** The primers used for PCR amplification in this study.

Primers	Targeted Genes	Sequence (5′-3′)
TL17	M13F	GTAAAACGACGGCCAG
TL18	M13R	CAGGAAACAGCTATGAC
TL19	*NEO* split F	GGGCGCCCGGTTCTTTTTGTCA
TL20	*NEO* split R	TTGGTGGTCGAATGGGCAGGTAGC
TL59	*NEO* R4	TGTGGATGCTGGCGGAGGATA
TL67	*STE20A* ⍺ F	CCAAAAGCTGATGCTGTGGA
TL68	*STE20A* a R	AGGACATCTATAGCAGAT
TL69	*STE20A* a F	TCCACTGGCAACCCTGCGAG
TL70	*STE20A* a R	ATCAGAGACAGAGGAGCAAGAC
TL217	*GAPDH* qRT-PCR F	TGAGAAGGACCCTGCCAACA
TL218	*GAPDH* qRT-PCR R	ACTCCGGCTTGTAGGCATCAA
TL235	*CMP1* KO F1	GGGGTAAAAGAGGGAGGATGAGAC
TL236	*CMP1* KO R1	CTGGCCGTCGTTTTACAGAACACCCGCCGCTGAACTTT
TL237	*CMP1* KO F2	GTCATAGCTGTTTCCTGCTCCGCTGCAACCAAGGCTACCA
TL238	*CMP1* KO R2	TGCGCGGCTCGAGACACAAGA
TL239	*CMP1* KO F3	TCCTCCGACTCGCGCCTCATCAG
TL240	*CMP1* KO R3	AGCTATCGCCGGCCCATTACCATC
TL241	*CMP1* KO F4	GCCCACGCGCCCACATACAT
TL481	*CMP1* PRO F1	ACGGTATCGATAAGCTTGACGGGTGCGGACGACATTTAGATTT (*Hind*III)
TL482	*CMP1* PRO R1	CTAGAACTAGTGGATCCACAGCATGCACACTCTCTGCAT (*BamH*I)
TL483	*CMP1-mCherry* F1	TTAGTAAACTCGCCCAACATGTCTGGATCCATGGCTGGCAGGTGGCCGCTGC (*BamH*I)
TL484	*CMP1-mCherry* R1	CTTGCTCACCATTCTAGAACTAGTGGATCCAAGCAAGACGGCAGCACCAAAC (*BamH*I)
TL535	*CMP1* QPCR F1	TCCTGGTATCTCCACCTCTTC
TL536	*CMP1* QPCR R1	CAAACAGCATGCCGACAAC
TL554	GFP-CMP1 F	GACGAGCTGTAcGGATCCATGGCTGGCAGGTGGCCGCTGCAC (*BamH*I)
TL555	GFP-CMP1 R	CTGGCGGCCGTTACTAGTTTAAAGCAAGACGGCAGCACCAAA (*Spe*I)
TL562	*CMP*1 Comp F	GATATCGAATTCCTGCAGCCCGGGGGATCCGCCCACGCGCCCACATACATCCTCGC (*BamH*I)
TL563	*CMP1* Comp R	CGGTGGCGGCCGCTCTAGAACTAGTGGATCCTGCGCGGCTCGAGACACAAGAGTAGA (*BamH*I)

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
