# Peer review of "A Predicted Mannoprotein Cmp1 Regulates Fungal Virulence in Cryptococcus neoformans"

_pathogens, 2020, doi:10.3390/pathogens9110881_

Round 1

Reviewer 1 Report

In the manuscript entitled,” A predicted mannoprotein Cmp1 regulates fungal virulence in Cryptococcus neoformans” The authors revealed that a predicted mannoprotein Cmp1 regulates fungal virulence in C. neoformans by functional analysis. They suggested that Cmp1 is required for the virulence of C. neoformans through capsule formation and virulence assay. Although cmp1 mutant does not have many phenotypes and it is already known that mannoprotein is involved in Cryptococcus virulence, it is interesting that virulence attenuates in both cases of CMP1 knockout and overexpress. Overall, this manuscript is simple and clear with the following exceptions:

Major:

  1. The authors showed that, unlike all other conditions, Cmp1 accumulates in the vacuole only in the capsule induction condition (Fig 2). Since GFP is stable and cells are cultured for a long time, partially degraded GFP-tagged Cmp1 may accumulate in the vacuole, so it is thought that more accurate information can be obtained by comparing the expression patterns of Cmp1 in nutrient condition and starvation condition.

  1. The authors showed that the capsule size was reduced in the cmp1 mutant as an important virulence factor (Fig 4A). In many other stress conditions such as melanin, starvation, high temperature, osmotic/oxidative stress, drugs, etc., there is no difference between wt and cmp1 mutant, and only the capsule size is slightly different. On the picture, the capsule size does not appear to have significantly decreased. For more accurate information, it is likely that the authors should perform capsule size comparison experiments in other capsule-inducing media such as diluted SAB media and RPMI-containing media.

Minor:

Line 113: H99 and KN99a à H99 (MATα) and KN99 (MATa)

Line 174: typo 37 C

Reviewer 2 Report

In this submission, Han et al. characterized the C. neoformans Cmp1 mannoprotein. The authors employed knockout, rescued and overexpression mutants to assay at which point of the fungal life cycle Cmp1 plays a role. It was found that Cmp1 may regulate capsule formation, and most importantly, has a notable role in infection.

Overall, the experiments are well-controlled and the manuscript is written in appropriate language. Although very interesting, the manuscript leaves the lingering question of just how Cmp1’s role in capsule formation relates to infection in a murine model of cryptococcosis. Since not much insight is presented in the discussion in that regard, it is important to provide experimental evidence to connect the two phenomena.

Major concerns:

  1. Several studies have shown that capsule formation (by bacteria and fungi) may affect complement killing and phagocytosis by phagocytes such as macrophages and neutrophils. In light of the data presented by Han and colleagues, it is important to show whether components of the host complement system damage C. neoformans. This can be done by incubating the authors’ Cmp1-WT and -mutant strains with mouse/human serum and then testing whether the fungi remain viable after such incubation.

  2. As stated by the authors, monocytes and macrophages are the first host cells that encounter C. neoformans. Since Cmp1 plays a role in capsule formation, it is plausible that Cmp1 modulates the expression of membrane components that facilitate recognition and ingestion by phagocytic cells. The authors should demonstrate this in vitro by showing whether macrophages (primary or cell lines) internalize Cmp1-WT and -mutant strains differently. Are the strains internalized at different rates and do they survive – within macrophages – at the same levels?

Round 2

Reviewer 1 Report

Accept the results of your revision experiment.

Reviewer 2 Report

Han et al. have addressed all of the reviewers’ suggestions.

This manuscript is a resubmission of an earlier submission. The following is a list of the peer review reports and author responses from that submission.